# Phosphogypsum-Based Ultra-Low Basicity Cementing Material

**DOI:** 10.3390/ma15196601

**Published:** 2022-09-23

**Authors:** Pengping Li, Xinxing Zhang, Mingfeng Zhong, Zhihong Fan, Jianbo Xiong, Zhijie Zhang

**Affiliations:** 1Key Laboratory of Harbor & Marine Structure Durability Technology, Ministry of Communications, Guangzhou 510230, China; 2School of Materials Science and Engineering, South China University of Technology, Guangzhou 511442, China

**Keywords:** low alkalinity, cementitious material, phosphogypsum, granulated ground blast slag, sulphoaluminate cement

## Abstract

Traditional Portland cement is widely used in the preparation of various hydraulic concrete. However, the high alkalinity produced by cement hydration threatens the survival of aquatic animals and plants. In this paper, a new eco-friendly, ultra-low alkalinity, cementitious material was prepared with industrial waste phosphogypsum, granulated ground blast slag (GGBS) and sulphoaluminate cement. When appropriate proportions are used, the pH value of the test blocks’ pore solutions at different ages were all less than 9, showing the remarkable characteristic of ultra-low alkalinity. The XRD and SEM analyses showed that the 56 d hydration products were mainly ettringite and hydrated calcium silicate, and the content of Ca(OH)_2_ was not detected. The new cementitious material also has the advantages of short setting time, low heat of hydration, high strength of cement mortar and the ability to fix harmful substances in phosphogypsum, such as phosphate, fluoride and Cr and Ba elements. It has a broad application prospect in the construction of island and reef construction, river restoration and so on.

## 1. Introduction

As Portland cement concrete is widely used in the construction of artificial coastal zones and islands, the pH value of pore solution of Portland cement concrete material is higher than 13, but most terrestrial and aquatic animals and plants live in neutral or weakly alkaline environments [1,2,3]. Concrete improves the infrastructure of nearshore and reef, but it also changes the living environment of living creatures, some of which may die due to these changes in the living environment [4,5,6,7,8]. In order to improve the biocompatibility of cement-based cementitious materials, it is necessary to reduce their basicity.

The conventional means of reducing alkali in concrete is to reduce the content of Ca(OH)_2_ crystals after cement hydration by adding supplementing cement materials (SCM) [9,10,11,12,13]. However, even if the dosage of fly ash [9,14] and silica fume [15,16,17] is increased to 50%, the pH of the pore solution is still above 12. It is difficult for ordinary Portland cement to directly prepare low alkalinity concrete.

At present, in addition to ordinary Portland cement, cementitious materials mainly include: aluminate cement, phosphate cement and sulphoaluminate cement. After hydration, the alkalinity in the pores of aluminate cement is 11.4~12.5 [18]. Although the alkalinity of phosphate cement pores after hydration is 7~9 [19], its strength will decrease when in contact with water for a long time, and the hydration of magnesium phosphate cement will produce ammonia and harm the environment, which does not meet the ecological requirements. The main hydration products of sulphoaluminate cement are ettringite (AFt) and a small amount of calcium silicate hydrate gels. As such, sulphoaluminate cement as a cementitious material has the potential of preparing low-basicity concrete.

Phosphogypsum, with gypsum as the main component, is the waste of the phosphorus chemical industry. Phosphogypsum is mainly composed of calcium sulfate dihydrate, but it also contains a small amount of impurities such as phosphoric acid, hydrofluoric acid and heavy metals [20,21]. Currently, China has 400 million tons of phosphogypsum in storage, and it is increasing at the rate of 50 million tons per year. However, the utilization rate is only 40% [22,23,24]. It is mainly used for cement retarder, gypsum board, building gypsum powder, etc. [25,26,27], but the consumption is small and the performance of phosphogypsum products is poor [28,29,30,31,32,33,34,35]. As a sulfate activator, the amount of gypsum used in the super sulfate cement (SSC) with granulated blast furnace slag as the main raw material can reach 10–20%, but the application of SSC is limited by its long setting time, low early strength [36,37,38] and other deficiencies.

Combined with the above factors, in this study, a new type of ecologically friendly cementite with normal setting time and ultra-low basicity was prepared by using phosphogypsum as the main raw material with an appropriate amount of high-activation S95 GGBS and a small amount of low-basicity sulphoaluminate cement. The setting time of the cementitious material, the strength of the cement mortar, the heat of hydration, the pH value of the pore solution, the toxicity leaching test and the type of hydration products were also tested and analyzed, and the possible hydration process was discussed by microscopic analysis and thermal analysis.

## 2. Experimental Materials and Methods

### 2.1. Experimental Materials

The raw materials in this study mainly consist of phosphogypsum, S95 granulated ground blast slag and sulphoaluminate cement, which are combined with Subert efficient water reducing agent for molding, curing and test characterization. Its main components and properties are as follows.

The raw materials are characterized by XRF, XRD and basic performance tests of cement, such as setting time, flexural strength and compressive strength. Table 1 shows the chemical composition of the three raw materials, Figure 1 shows XRD patterns of them.

Sulphoaluminate cement minerals mainly include Ca_4_(Al_6_O_12_)SO_4_, C_2_S, SiO_2_, Al_2_O_3_ and so on. They have short setting times and high early strengths. The 3 d compressive strength of sulphoaluminate cement reaches 42.5 Mpa, and its performance test results is showed in Table 2. Phosphogypsum consists of a large amount of CaSO_4_·2H_2_O and a small amount of impurities such as SiO_2_, phosphate and fluoride, which are harmful to the environment and difficult to treat [39]. The XRD pattern of granulated ground blast slag is steamed bun peak, indicating that it is mainly composed of glass phase. The 28 d activity index of granulated ground blast slag was 102% showed by Table 3. The water reducing agent was polycarboxylic acid high-efficiency compound admixture, and the water reducing efficiency was 25%. It is a polyether-type C6 mother liquor, mainly composed of APEG, TPEG and polyether with maleic anhydride as the main chain branched with different side chain lengths. The solids content is 50%, its performance is showed in Table 4. It has good dispersibility and water reduction without retardation.

### 2.2. Experimental Methods

The specific mixing ratio of compound cementitious system is shown in Table 5:

After the experimental materials were accurately weighed according to proportions in Table 5 and molded following the molding method specified in GB17671-1999 “Cement Mortar Strength Test Method”, the specimen could be normally removed after 24 h curing in a standard curing room with a temperature of 20 ± 2 °C and relative humidity greater than 95%. There were three samples in each group per age. The BG group refers to the blank control group containing 100% sulfoaluminate cement.

The flexural and compressive strength of the samples were tested by SUN’s microcomputer-controlled electro-hydraulic servo pressure testing machine TYE-300 and TYE-300 Cement flexural and compressive testing software. The mortar strength test was carried out in accordance with the GB/T 17671-2021 standard, and the test temperature was 20 ± 1 °C. Testing was performed using a 40 mm × 40 mm × 160 mm prism. Three identical samples of each formulation were tested for flexural strength. After they were broken, six samples were tested for compressive strength, and the final results were averaged [40,41,42].

The XRD and SEM tests were carried out after the hydration was stopped at a certain age. X ‘Pert PRO X-ray instrument from PANalytical Company of Holland is used for the XRD test. Cu target (λ = 1.5418 A), tube pressure and tube flow were 40 KV and 40 mA. Nova Nano SEM450 high-resolution scanning electron microscopy (SEM) was used to analyze the micromorphology of the samples. The acceleration voltage was 5–20 kV, and scanning depth was about 5-10 μm. TG-DSC simultaneous thermal analysis (SiC furnace body) was used to analyze the changes in raw materials and hydration products. The toxicity leaching experiment was based on HJT 299-2007 standard. The total phosphorus and fluoride were detected by ammonium molybdate spectrophotometry and ion chromatography, respectively; total chromium and total barium concentrations were detected by inductively coupled plasma atomic emission spectrometry, and total arsenic was determined by atomic fluorescence method. The heat of hydration was tested by adiabatic method from GB/T12959-2008.

The pH value of pore solution test: samples reached certain ages are broken into squares and soaked in anhydrous ethanol in less than 5 °C freezer to stop the hydration at 24 h, put in a vacuum dryer to dry for 24 h, and then ground and sieved through the 0.08 mm sieve. Three small samples were weighed and stirred with water at the ratio of solid to solution 1:10. The mixtures precipitated for 6 h, and the pH value of the supernatant was tested with a pH meter.

## 3. Experimental Results and Discussion

### 3.1. Setting Time of the Slurry

The initial and final setting time of the slurry was determined by the standard determination method of the Vicat apparatus (GBT1346-2011). The blank group (BG) is sulphoaluminate cement. As shown in Figure 2, it can be seen that the addition of phosphogypsum and GGBS prolongs the initial and final setting time. When the water–binder ratio increased by 0.1, the initial setting time increased by 8–20 mins and the final setting time increased by 31–34 mins. When the water–binder ratio was fixed, the more sulphoaluminate cement content, the shorter the setting time. In general, the setting time of the system was similar to that of Portland cement, which meets the actual production demand.

### 3.2. Mortar Strength Test

According to Table 6, the 3 d compressive strength (18.6 MPa) of No.2 was much higher than that of super sulfate cement [43,44]: the 28 d and 56 d compressive strength reached 27.6 MPa and 29.5 MPa, respectively. This means that this kind of cementitious material has the potential to be used as an ordinary concrete cementitious material. However, when the water–binder ratio increases to 0.6, the strength of each group of cementitious materials at the same hydration age decreases significantly, indicating that great attention should be paid to controlling the water–binder ratio when applying this kind of cementitious material.

### 3.3. Pore Solution pH Value

It can be seen from Figure 3 that the pH value of the pore solution in each group decreases with the extension of hydration time and increases with the increase in the water–binder ratio. When the water–binder ratio was 0.5, the pH value of the pore solution at the age of 28 days was lower than 9.0, and the sample of No.2 had the lowest pH value and the highest compressive strength at the same time. The lower pH value of the pore solution makes this new cementitious material more conducive to plant growth and ecological regulation than Portland cement [45].

The pH value of the pore solution will increase with the increase in the water–binder ratio. The lower the water–binder ratio, the lower the porosity, and the dissolved Ca(OH)_2_ in the pore solution also decreases [46], so its pore solution pH value was lower.

### 3.4. XRD Analysis of Hydration Products

The XRD patterns of No.2 showed that the hydration product was mainly composed of Ettringite (AFt), C-S-H and gypsum; calcium hydroxide and calcium sulphoaluminate (C_4_A_3_$\bar{S}$) were not detected.

As can be seen from Figure 4, with the increase in curing age, the diffraction peaks of AFt and C-S-H become higher, indicating the formation of more hydration products. Unreacted gypsum was detected in the samples of each age, which may have an adverse impact on the water erosion resistance of the material. Therefore, it is necessary to continue to optimize the proportion in the future to ensure the long-term performance of the cementitious material.

### 3.5. SEM Analysis of Hydration Products

As can be seen from Figure 5, the phosphogypsum gradually decreased, while AFt and C-S-H gels in hydration products increased, and the cement block gradually densified from 3 d to 56 d. The hydration products of 3 d old samples were mainly composed of AFt and a small amount of C-S-H gel minerals. As shown in Figure 6, AFt was a needle-stick crystal and its amount was small, and hydration products of 3 d old samples did not lap well and were not dense enough, and a lot of phosphogypsum can still be observed. With the increase in hydration age, more AFt was continuously generated, and more C-S-H gels were generated, increasing the density of the system, and the denser structure could be seen in the 56 d old sample. Hexagonal sheet Ca(OH)_2_ crystals were not observed in the samples of various ages. Meanwhile, it can be seen from the SEM images that the amount of phosphogypsum kept decreasing.

### 3.6. Heat of Hydration Analysis

As can be seen from Figure 7, the overall hydration heat of the system is much lower than sulfoaluminate cement (BG). The first 40 h is the main hydration exothermic time of the rubber material, the hydration rate is the fastest within 24 h, and the effect reaches the extreme in 8–10 h. The temperature rise of No.2 is 3.98 °C, and that of BG is 20.73 °C; the 7 d hydration heat of No.2 is 56% lower than that of BG. The early hydration rate of No.3 is fast due to the high content of sulfoaluminate cement, but its late strength is lower than that of No.2, indicating that this ratio cannot better stimulate the reaction between phosphogypsum and GGBS.

The new cementitious material has very low hydration heat and temperature rise during reaction, which meets the temperature rise conditions of mass concrete projects

### 3.7. TG-DSC Analysis

As shown in Figure 8, the peak near 120 °C was formed by the dehydration of AFt, and the peak value became larger with the increase in age and was the largest at 56 d, indicating that AFt continued to increase at each age. The peak near 150 °C [47] is formed by the dehydration of C-S-H gels. This peak is not obvious at 3 d and 7 d but is obvious at 28 d and 56 d, indicating that C-S-H is still generated in the later stage of hydration. The peak of 245 °C [48] in the figures of 3 d and 7 d was formed by the dehydration of Al(OH)_3_, while it did not appear in the other two ages, indicating that Al(OH)_3_, as an intermediate product, participated in the hydration process. The peak of 728 °C [49] is formed by the dehydration of CaCO_3_, and this peak in the four ages indicates that a small amount of carbonization will occur in the system. The decomposition temperature of Ca(OH)_2_ [47] is around 600 °C, there is only a slight change in the DSC curve, and it is known that its content is too small to be observed.

The results of simultaneous thermal analysis can verify the analyses of XRD and SEM. From these analyses, the progress of hydration can be known. It is also verified that the cementitious material can continue the hydration reaction in a low-alkalinity environment.

### 3.8. Toxicity Leaching Test Analysis

The data from Table 7 show that the cementing material can fix a large amount of fluoride and inorganic phosphate after curing for 3 d, and both are below the standard limit. Although Ba and Cr are present in small amounts, they can be immobilized in the cementitious material. At 56 d, As was still only 29% fixed. As the curing age increases, the amount of re-curing decreases.

Compared with the original phosphogypsum, the leaching ion concentrations of the tested phosphogypsum fillings were significantly lower, and they were all lower than the concentration limits of hazardous components in the identification standard for hazardous wastes-leaching toxicity identification (GB5085.3-2007), indicating that the phosphogypsum fillings were lower than the concentration limits of hazardous components. Contributes to the curing of toxic and harmful ions.

## 4. Reaction Mechanism Analysis

According to the previous test results, the reaction mechanism is speculated. The Monosulfide hydrated calcium sulphoaluminate from the direct hydration of C_4_A_3_$\bar{S}$ can react with gypsum(C$\bar{S}$H_2_) and H_2_O(H) to form ettringite and provide early strength, and its setting time was similar to that of Portland cement, which overcame the disadvantages of low early strength and long setting time of super sulfated cement.
(1)$$C_{4}A_{3}\bar{S}+18H\;\;\to \;\;C_{3}A\cdot C\bar{S}H_{12}\;+2\mathrm{AH}$$
(2)$$C_{3}A\cdot C\bar{S}H_{12}\;+2C\bar{S}H_{2}\;+16H\;\;\to \;\;C_{6}A{\bar{S}}_{3}H_{32}$$

C_2_S in sulfoaluminate cement continuously slowly hydrates to produce C-S-H and calcium hydroxide (CH), which is absorbed in various reactions. With alkali excitation, phosphogypsum continues to be consumed by hydration.
C_2_S + 2H  →  C-S-H + CH(3)
CH + AH + nH  →  CAH_10_(4)
(5)$$\mathrm{CH}+{\mathrm{CAH}}_{10}\;+3C\bar{S}H_{2}\;+15H\;\;\to \;\;C_{6}A{\bar{S}}_{3}H_{32}$$

The active Si^4+^ and Al^3+^ in GGBS continued to react with Ca(OH)_2_ intermediates, which reduced the alkalinity of the system and ensured the sustained growth of the later strength.
CH + Si^4+^  →  C-S-H(6)
CH + Al^3+^  →  C-A-H(7)
(8)$$C-A-H+C\bar{S}H_{2}\;+\mathrm{nH}\;\;\to \;\;C_{6}A{\bar{S}}_{3}H_{32}$$

CaSO_4_ in phosphogypsum participated in secondary hydration to generate AFt. Gypsum provided a large amount of SO_4_^2−^ to promote the hydration reaction (8) and promoted the decomposition of the GGBS.

In addition, the neutralization of CaF_2_ and Ca_3_(PO_4_)_2_ with a small amount of acid in phosphogypsum would help reduce the alkalinity of the system but did not affect the strength [50].

AFt generated by hydration of C_4_A_3_$\bar{S}$ in sulphoaluminate cement provided the early strength. Continuous secondary hydration not only reduces the alkalinity of the system by consuming the hydration intermediate CH but also provides a later increase in strength. Phosphogypsum can not only activate the activity of slag as an activator but also participate in the hydration reaction at all stages.

## 5. Conclusions

The ultra-low-alkalinity cementitious material with normal setting time was successfully prepared with phosphogypsum as the main raw material. Its 7 d hydration heat was only 87.74 J/g, much lower than that of sulphoaluminate cement. The hydration product was mainly composed of AFt and C-S-H gel, the pH value of 28 d pore solution was 8.2. The 28 d flexural strength and compressive strength of cement mortar reached 4.2 MPa and 27.6 MPa, respectively, and the toxic ions in phosphogypsum are fixed well. This new type of cementitious material can not only reuse a large amount of industrial waste, but it also has eco-friendly properties. It may be applied in the construction of islands and reefs, river restoration and so on. It should be noted that the effect of unreacted gypsum in the material on water resistance needs to be further studied.

## Figures and Tables

**Figure 1 materials-15-06601-f001:**
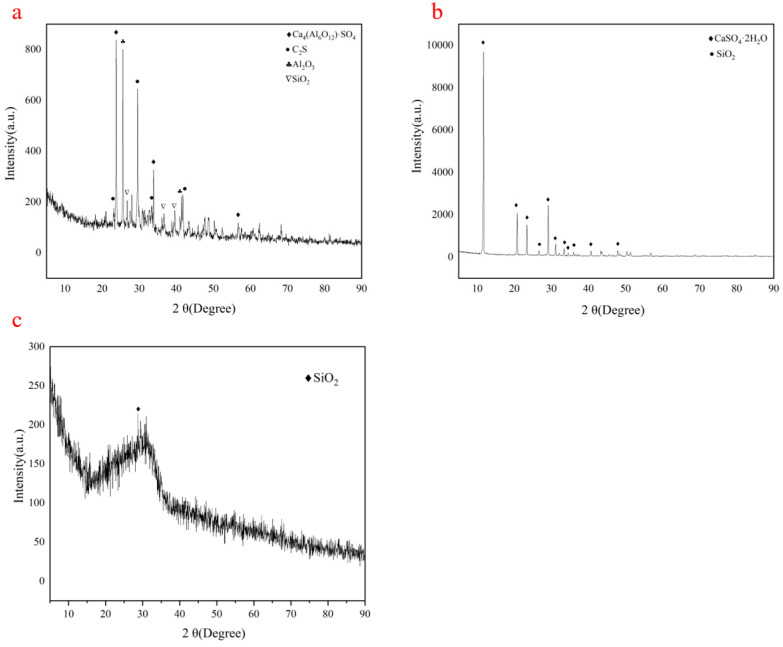
XRD patterns of sulphoaluminate cement (**a**), phosphogypsum (**b**) and granulated ground blast slag (**c**).

**Figure 2 materials-15-06601-f002:**
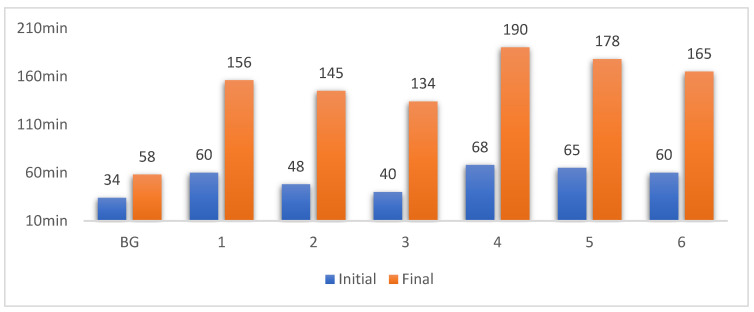
The initial and final setting time of the slurry.

**Figure 3 materials-15-06601-f003:**
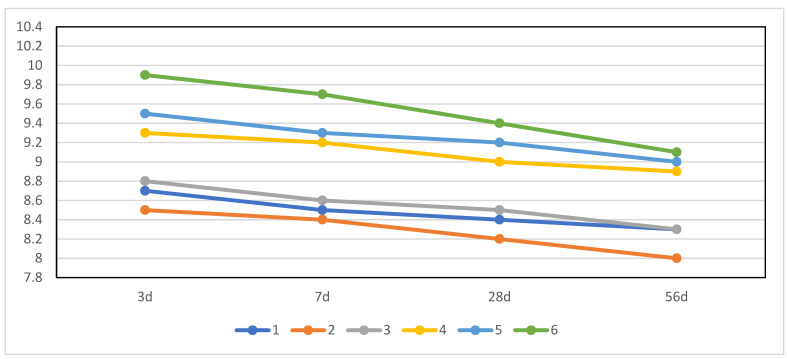
The pH value of pore solution at various hydration ages.

**Figure 4 materials-15-06601-f004:**
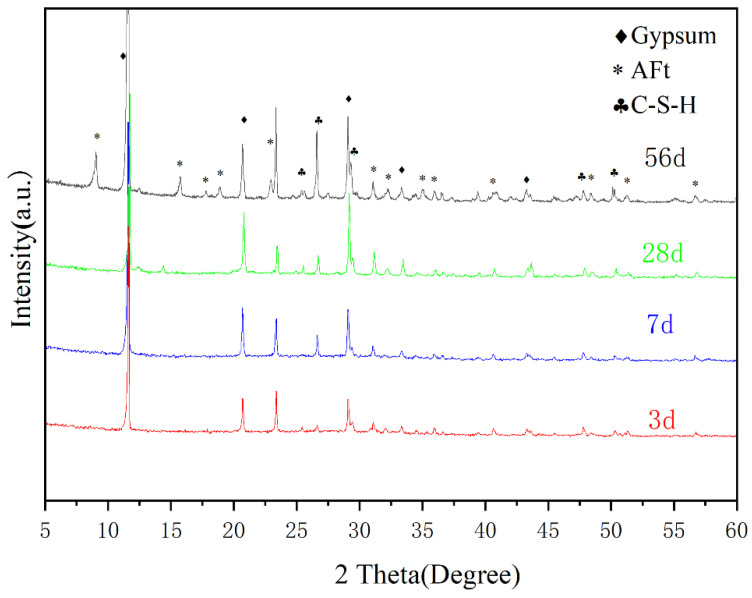
XRD patterns of hydration products of No.2 at different ages.

**Figure 5 materials-15-06601-f005:**
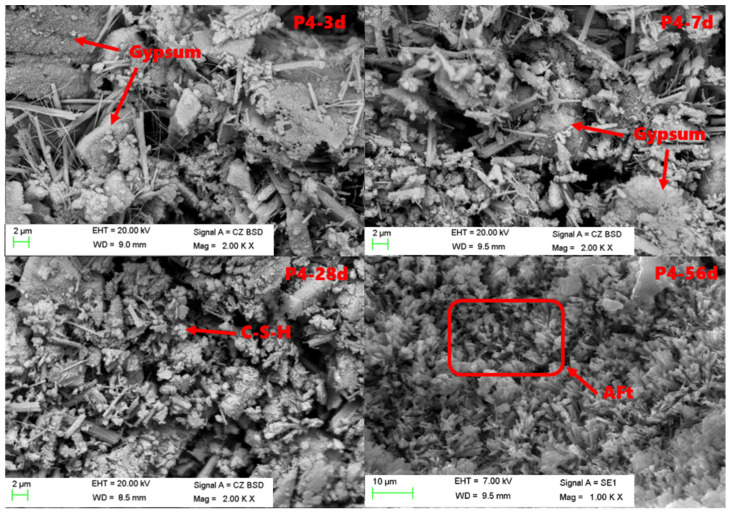
SEM images of hydration products of No.2 at various ages.

**Figure 6 materials-15-06601-f006:**
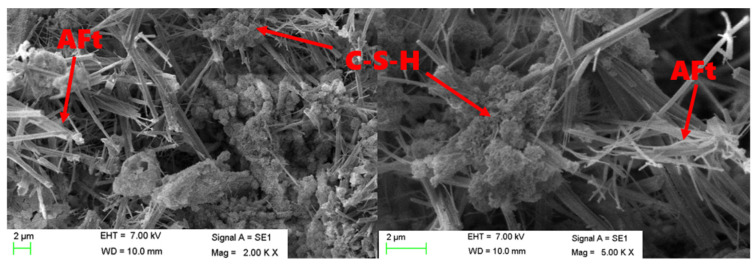
SEM images of hydration products of No.2 at 28 d age.

**Figure 7 materials-15-06601-f007:**
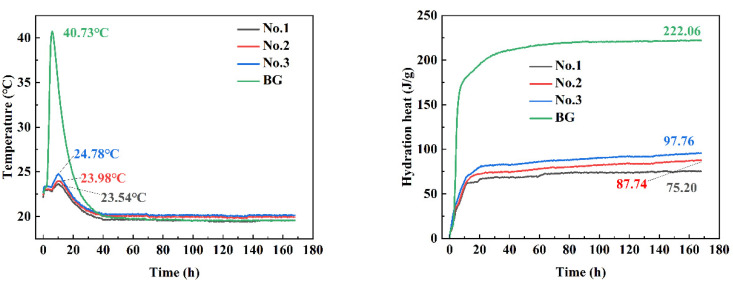
Temperature–time curve (**left**); heat of hydration curve (**right**).

**Figure 8 materials-15-06601-f008:**
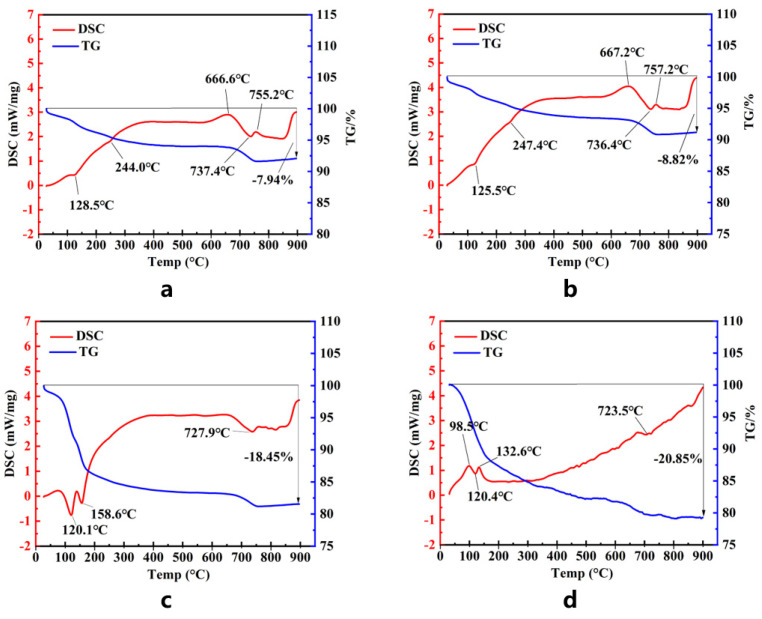
TG-DSC test results of 3 d (**a**), 7 d (**b**), 28 d (**c**), 56 d (**d**).

**Table 1 materials-15-06601-t001:** Raw material composition (wt%).

	CaO	SO_3_	SiO_2_	Al_2_O_3_	Fe_2_O_3_	MgO	P_2_O_5_	K_2_O	F
phosphogypsum	36.05	52.35	8.83	0.17	1.56	-	1.18	0.27	0.71
S95 GGBS	40.56	-	33.89	14.47	0.36	7.43	-	-	-
Sulphoaluminate cement	44.63	8.64	23.68	31.28	1.65	1.36	-	1.6	-

**Table 2 materials-15-06601-t002:** Performance test results of sulphoaluminate cement.

Specific Surface Area (m^2^/kg)	Setting Time/ min		Flexural Strength/MPa		Compressive Strength/MPa		pH
	Initial	Final	3 d	28 d	3 d	28 d	
380	34	58	4.6	7.3	41.0	53.1	11.20

**Table 3 materials-15-06601-t003:** Granulated ground blast slag performance test results.

Density/(g/cm^3^)	Specific Surface Area/(m^2^/kg)	The Activation Index/%		Fluidity/%
		7 d	28 d	
≥2.8	429	90	102	≥95

**Table 4 materials-15-06601-t004:** Performance index of polycarboxylic acid high-efficiency compound admixture.

Formaldehyde Content/%	Net Slurry Fluidity/mm	Sodium Sulfate Content/%	Chloride Ion Content/%	Alkali Content/%
0.003	280	1.0	0.01	1.1

**Table 5 materials-15-06601-t005:** The mixing ratio of the mortar experiment.

NO.	Sulphoaluminate Cement/wt%	S95 GGBS/wt%	Phosphogypsum/wt%	Water–Binder Ratio	Water Reducing Agent/wt%
1	15.0	35.0	50.0	0.5	0.5
2	20.0	30.0	50.0	0.5	0.5
3	25.0	25.0	50.0	0.5	0.5
4	15.0	35.0	50.0	0.6	0.5
5	20.0	30.0	50.0	0.6	0.5
6	25.0	25.0	50.0	0.6	0.5
BG	100.0	-	-	0.5	0.5

**Table 6 materials-15-06601-t006:** Strength of test blocks at various ages.

No.	Strength of 3 d (MPa)		Strength of 7 d (MPa)		Strength of 28 d (MPa)		Strength of 56 d (MPa)	
	Flexural	Compressive	Flexural	Compressive	Flexural	Compressive	Flexural	Compressive
1	2.6	17.8	3.7	21.4	4.0	22.3	4.2	23.5
2	2.9	18.6	4.1	25.1	4.2	27.6	4.6	29.5
3	2.8	18.2	4.0	24.7	4.3	26.9	4.5	27.6
4	1.4	8.9	1.6	11.3	2.2	13.4	2.3	14.2
5	1.5	9.7	2.2	11.9	2.6	14.7	2.7	16.4
6	1.7	11.3	2.3	13.3	2.7	16.8	2.9	18.5
BG	4.6	41.0	5.6	47.6	7.3	53.1	7.9	56.4

**Table 7 materials-15-06601-t007:** Leaching toxicity test results.

Type of Toxic Ion	Control Group/(mg·kg^−1^)	Concentration of Toxic Ions of 3 d/(mg·kg^−1^)	Concentration of Toxic Ions of 56 d/(mg·kg^−1^)	Reference Standard of Mass Concentration Detection Limit/(mg·kg^−1^)
				GB 5085.3-2007
Total Ba	0.711	0.067	0.053	0.4≤
Total Cr	0.752	0.191	0.125	0.4≤
Inorganic fluoride	87.925	9.021	8.125	100≤
Total As	9.704	7.924	6.851	15≤
Inorganic phosphate	1.728	0.155	0.107	-

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
