# Peer review of "Phosphogypsum-Based Ultra-Low Basicity Cementing Material"

_materials, 2022, doi:10.3390/ma15196601_

Round 1
Reviewer 1 Report
Attached document with reviewer comments.

Author Response
Dear Editor,
Thank you for your kind letter which about our article “Ultra-low basicity eco-friendly cementitious material with low heat of Hydration (Manuscript ID: materials-1903970)” on September 8th,2022. We revised the manuscript in accordance with the reviewers' comments, and carefully proof-read the manuscript to minimize typographical, grammatical, and bibliographical errors.
Here below is our description on revision according to the reviewers' comments.
1.The reviewer's comment: The reference number 30 should be reviewed.
The authors' Answer: This is a Chinese reference, its DOI:10.27393/d.cnki.gxazu.2019.000525.
2.The review's comment: The lack of information methodology or the standards used to reproduce the tests for the raw materials.
The authors' Answer: The information methodology of the raw materials is added to the beginning of Chapter 2.1.
3.The reviewer's comment: Number of samples from different test are not presented.
The authors' Answer: The number of samples is added to the second paragraph of Chapter 2.1
4.The review's comment: Table 5 no present the BG experimental values.
The authors' Answer: Group BG is a blank control group whose strength test results have been supplemented to Table 6, it consists of 100% sulfoaluminate cement.
5.The review's comment: The figure 3 should be improved perhaps by removing the test values from the graph.
The authors' Answer: The test values ​​of figure 3 have been removed.
6.The review's comment: Below the figure 6 there are the paragraph “The temperature rise of Group2 is 3.98℃, and that of BG is 20.73℃; the 7d hydration heat of Group2 is 56% lower than that of KB”. The authors should explain the meaning of KB acronym.
The authors' Answer: This is a typo, the KB here should be BG, and it has been corrected.
7.The review's comment: In the conclusion chapter the authors should explain that a certain amount of unreacted gypsum was detected, and its possible effect on the water erosion resistance of the material.
The authors' Answer: Conclusions have been optimized and an explanation of gypsum residues and their effects has been added.
Many grammatical or typographical errors have been revised.
All the lines and pages indicated above are in the revised manuscript.
Thank you and all the reviewers for the kind advice.
Sincerely yours,
Xinxing Zhang

Reviewer 2 Report
The paper presents an interesting approach based on the Ultra-low basicity eco-friendly cementitious material with low heat of hydration. However, the innovation of the current research work should be further highlighted and emphasized. At the same time, the authors should consider the following comments to greatly improve the quality of the paper.
1. Kindly consider editing the title, as it doesn't perfectly fit with the proposed idea.
2. In the abstract, an abbreviation GGBS appeared in a larger font than the rest of text which needs to be unified. The final statement in abstract is too generic and needs to be rewritten to be more concise and accurate.
3. The introduction needs to be improved by relating to the mechanics of the studied materials and their mechanical characteristics. The references to be included are: 10.1016/j.compstruct.2021.114698,
10.1016/j.porgcoat.2022.107015, 10.1016/j.jiec.2022.06.023
4. The main physcial and chemical properties of the Subert efficient water has to be listed in the raw materials table.
5. Under which standard were the mechanical tests performed? For each applied test (especially the detailed ones such as flexural and compression), the sample geometry has to be identified, the testing procedure has to be mentioned and the number of samples per configuration has to be declared.
6. For table 6, where are the variance in the data set presented? How many repetitions for each value were taken intp consideration?
7. For SEM analysis, what was the SEM parameters applied (accelerating voltage and working depth)?. Also, the scale is missing in the image. Are there different variations for other spots to validate the presented idea in your SEM analysis?
8. The conclusion needs to be modified to summarize the research outcomes in short statements with clear observations.
Author Response
Dear Editor,
Thank you for your kind letter which about our article “Ultra-low basicity eco-friendly cementitious material with low heat of Hydration (Manuscript ID: materials-1903970)” on September 8th,2022. We revised the manuscript in accordance with the reviewers' comments, and carefully proof-read the manuscript to minimize typographical, grammatical, and bibliographical errors.
Here below is our description on revision according to the reviewers' comments.
1.The reviewer's comment: Kindly consider editing the title, as it doesn't perfectly fit with the proposed idea.
The authors’ Answer:We removed “eco-friendly” and re-titled “Phosphogypsum based ultra-low basicity cementing material”.
2.The review's comment:In the abstract, an abbreviation GGBS appeared in a larger font than the rest of text which needs to be unified. The final statement in abstract is too generic and needs to be rewritten to be more concise and accurate.
The authors’ Answer:The font size has been adjusted and the final statement in abstract has been simplified.
3.The review's comment:The introduction needs to be improved by relating to the mechanics of the studied materials and their mechanical characteristics.
The authors’ Answer:We have added these three inferences, numbered as[41-43].
4.The review's comment:The main physcial and chemical properties of the Subert efficient water has to be listed in the raw materials table.
The authors’ Answer:Subert is the largest admixture manufacturer in China. We have not been able to test its specific chemical composition, but other properties have been added at the end of the third paragraph of Section 2.1.
5.The review's comment:Under which standard were the mechanical tests performed? For each applied test (especially the detailed ones such as flexural and compression), the sample geometry has to be identified, the testing procedure has to be mentioned and the number of samples per configuration has to be declared.
The authors’ Answer:We have supplemented the relevant standards and methods in the third paragraph of Section 2.2.
6.The review's comment:For table 6, where are the variance in the data set presented? How many repetitions for each value were taken into consideration?
The authors’ Answer:The flexural strength is from three repeated tests, and the compressive strength is from six tests, and the final results are averaged.
7.The review's comment:For SEM analysis, what was the SEM parameters applied (accelerating voltage and working depth)?. Also, the scale is missing in the image. Are there different variations for other spots to validate the presented idea in your SEM analysis?
The authors’ Answer:The acceleration voltage is 5-20kV, scanning depth is about 5-10μm. There is a scale bar in the lower left corner of each SEM image. XRD test and TG-DSC test can supplement to prove the existence and distribution of AFt, C-S-H gels and gypsum. At the same time, we supplemented two detailed SEM photos of 28d age.
8.The review's comment:The conclusion needs to be modified to summarize the research outcomes in short statements with clear observations.
The authors’ Answer:We have optimized the conclusion.
Many grammatical or typographical errors have been revised.
All the lines and pages indicated above are in the revised manuscript.
Thank you and all the reviewers for the kind advice.
Sincerely yours,
Xinxing Zhang

Reviewer 3 Report
Review Assignment for Materials-1903970:
In this work industrial waste phosphogypsum, granulated ground blast slag, and sulphoaluminate cement were employed to modify cementitious material features including alkalinity. Authors have considered the essential parameters and I see very interesting results in this paper. I hope the authors apply this revision on their paper since I see good potential in this paper to be published after revisions. Here are my comments:
- In the introduction, it would be good if the authors first mention concrete can be improved by supplementary cementitious materials (SCMs) such as silica fume, zeolite, or titanium dioxide. Some references to consider are: "Strength optimization of cementitious composites reinforced by carbon nanotubes and Titania nanoparticles", and "Investigation on optimal lightweight expanded clay aggregate concrete at high temperature based on deep neural network."
- In the introduction, “The conventional means of reducing alkali in concrete is to reduce the content of Ca(OH)2 crystals after cement hydration by using pozzolanic ash reaction.” Please use the most recent references about pozzolanic reaction. The following references are recommended: "Cement Paste Modified by Nano-Montmorillonite and Carbon Nanotubes", and "Multi-Objective Optimization of Sustainable Concrete Containing Fly Ash Based on Environmental and Mechanical Considerations."
- Please improve the quality of figures (e.g. Fig. 1) and the information in the figures should be more legible when the page is viewed at 100%.
- Page 4, Table 5, "BG" needs to be defined.
- Page 8, the texts in Fig. 5 are not clear. A different color or bigger font can be helpful.
- How did you identify the C-S-H gels in Fig. 5?
- Page 10, "The peak near 120℃ was formed by dehydration of AFt, and the peak value became larger with the increase of age, and was the largest at 56d, indicating that AFt continued to increase at each age. The peak near 150℃ is formed by dehydration of C-S-H gels. This peak is not obvious in 3d and 7d, but obvious in 28d and 56d, indicating that C-S-H is still generated in the later stage of hydration. The peak of 245℃ in the figures of 3d and 7d was formed by the dehydration of Al(OH)3, while it did not appear in the other two ages, indicating that Al(OH)3, as an intermediate product, participated in the hydration process. The peak of 728℃ is formed by the dehydration of CaCO3, and this peak in the four ages indicates that a small amount of carbonization will occur in the system. The decomposition temperature of Ca(OH)2 is around 600℃, there is only a slight change in the DSC curve, and it is known that its content is too small to be observed." It is highly recommended that add references for each peak. This reference may be useful: "Microstructural Characterization and Mechanical Properties of Cementitious Mortar Containing Montmorillonite Nanoparticles."
- The reference list should be revised (e.g. Ref. #20)
In the present form of the paper, my decision is a major revision. However, I would be glad to review the article after the authors’ revision.
Author Response
Dear Editor,
Thank you for your kind letter which about our article “Ultra-low basicity eco-friendly cementitious material with low heat of Hydration (Manuscript ID: materials-1903970)” on September 8th,2022. We revised the manuscript in accordance with the reviewers' comments, and carefully proof-read the manuscript to minimize typographical, grammatical, and bibliographical errors.
Here below is our description on revision according to the reviewers' comments.
1.The review's comment:In the introduction, it would be good if the authors first mention concrete can be improved by supplementary cementitious materials (SCMs) such as silica fume, zeolite, or titanium dioxide. Some references to consider are: "Strength optimization of cementitious composites reinforced by carbon nanotubes and Titania nanoparticles", and "Investigation on optimal lightweight expanded clay aggregate concrete at high temperature based on deep neural network."
The authors’ Answer:The relevant content has been supplemented in the Introduction, and references are also cited,numbered as [12,13].
2.The review's comment:In the introduction, “The conventional means of reducing alkali in concrete is to reduce the content of Ca(OH)2 crystals after cement hydration by using pozzolanic ash reaction.” Please use the most recent references about pozzolanic reaction. The following references are recommended: "Cement Paste Modified by Nano-Montmorillonite and Carbon Nanotubes", and "Multi-Objective Optimization of Sustainable Concrete Containing Fly Ash Based on Environmental and Mechanical Considerations."
The authors’ Answer:The relevant reference has been cited, number [9-11].
3.The review's comment:Please improve the quality of figures (e.g. Fig. 1) and the information in the figures should be more legible when the page is viewed at 100%.
The authors’ Answer:Figure 1 has been replaced with a cleaner version.
4.The review's comment:Page 4, Table 5, "BG" needs to be defined.
The authors’ Answer:Group BG is a blank control group whose strength test results have been supplemented to Table 6, it consists of 100% sulfoaluminate cement. It is explained in the second paragraph of Section 2.2.
5.The review's comment:5. Page 8, the texts in Fig. 5 are not clear. A different color or bigger font can be helpful.
The authors’ Answer:We tried different colors and none of the reds were noticeable, so we increased the font size and darkened the color. Now it becomes clearer.
6.The review's comment:How did you identify the C-S-H gels in Fig. 5?
The authors’ Answer:Through the analysis of the shape, we concluded that it is C-S-H. At the same time, we supplemented two detailed SEM photos of 28d age.
7.The review's comment:Page 10, "The peak near 120℃ was formed by dehydration of AFt, and the peak value became larger with the increase of age, and was the largest at 56d, indicating that AFt continued to increase at each age. The peak near 150℃ is formed by dehydration of C-S-H gels. This peak is not obvious in 3d and 7d, but obvious in 28d and 56d, indicating that C-S-H is still generated in the later stage of hydration. The peak of 245℃ in the figures of 3d and 7d was formed by the dehydration of Al(OH)3, while it did not appear in the other two ages, indicating that Al(OH)3, as an intermediate product, participated in the hydration process. The peak of 728℃ is formed by the dehydration of CaCO3, and this peak in the four ages indicates that a small amount of carbonization will occur in the system. The decomposition temperature of Ca(OH)2 is around 600℃, there is only a slight change in the DSC curve, and it is known that its content is too small to be observed." It is highly recommended that add references for each peak. This reference may be useful: "Microstructural Characterization and Mechanical Properties of Cementitious Mortar Containing Montmorillonite Nanoparticles."
The authors’ Answer:We have added references for each peak to the article,numbered as [48-50].
8.The review's comment:The reference list should be revised (e.g. Ref. #20)
The authors’ Answer:We have refined the references in detail.
Many grammatical or typographical errors have been revised.
All the lines and pages indicated above are in the revised manuscript.
Thank you and all the reviewers for the kind advice.
Sincerely yours,
Xinxing Zhang

Round 2
Reviewer 2 Report
The paper can be accepted.
Reviewer 3 Report
Review assignment for Materials-1903970:
I am satisfied with the changes in the revised manuscript. I recommend the paper to be published.